# Overview of Self-Management Skills and Associated Assessment Tools for Children with Inflammatory Bowel Disease

Angharad Vernon-Roberts , Richard B. Gearry and Andrew S. Day *

Department of Paediatrics, University of Otago, Christchurch 8011, New Zealand;
angharad.hurley@otago.ac.nz (A.V.-R.); richard.gearry@cdhb.health.nz (R.B.G.)
* Correspondence: Andrew.day@otago.ac.nz

**Abstract:** Self-management is a multi-modal approach for managing chronic conditions that encompasses a number of different elements; knowledge, adherence, self-regulation, communication, and cognitive factors. Self-management has been shown to be beneficial for adults with inflammatory bowel disease (IBD), and for children with IBD it may help them learn to take control of their complex treatment regimens and lead to positive disease outcomes. The development of self-management skills for children with IBD is vital in order to maximize their potential for health autonomy, but it is still an emergent field in this population. This review provides an over-arching view of the self-management elements specific to children with IBD, and highlights outcome measures that may be used to assess skills within each field as well as the efficacy of targeted interventions.

**Keywords:** self-management; inflammatory bowel disease; knowledge; adherence; communication; self-monitoring; self-efficacy; patient activation

## 1. Introduction

Inflammatory bowel disease (IBD) is a collective term for the clinical sub-types of Crohn's disease (CD) and ulcerative colitis (UC), both of which are complex, relapsing conditions affecting the gastrointestinal tract. The global incidence of pediatric IBD is increasing and it is now considered to be one of the most important and serious chronic diseases of childhood [1–5].

When IBD is diagnosed during childhood, it can be associated with more extensive disease, higher disease activity, and a more complicated progression than when diagnosed as an adult [6,7]. Treatment regimens may be complex involving the use of drug therapy, nutrition, surgical intervention, multidisciplinary team (MDT) involvement, and psycho-social input. It is, therefore, vital to identify an integrative approach that addresses all of these factors simultaneously, while being cognizant of the implication that children and their families will also be responsible for adhering to these multi-component strategies [8–11].

One such multi-modal approach is self-management, which involves children learning to take control of their own treatment strategies and gain those skills and attributes necessary to self-manage their IBD independently [12,13]. Self-management is centered on the concept of health autonomy, whereby the child's increasing age sees a paradigm shift of responsibility from disease management by their parent or family, to that of the individual. Research into the benefits of self-management for children with other chronic conditions has shown positive effects on clinical, behavioral, and psychological outcomes, with evidence of improvements to items such as; self-reported health status, health outcomes, school absenteeism, recreational activities, knowledge, quality of life, health care utilization, and self-efficacy [14,15]. In addition, self-management interventions have shown benefit among adults with IBD [16–20], but is an emerging field of study among children with IBD. While self-management interventions developed for adults with IBD may share components that are applicable to children, the pediatric population have unique characteristics

that need to be considered [21]. Their developmental and cognitive attributes, as well as their increasing age and development of health autonomy, require interventions and assessment tools to be tailored specifically to these factors. Children of different ages will have varying levels of understanding and literacy, and there is also a paradigm shift of responsibility from parents to their child as determined by age and cognitive developmental ability. Both of these factors dictate that the approach to pediatric self-management needs to be age and developmentally appropriate, a factor that will most likely not have been considered in adult targeted interventions or assessments.

Self-management is a critical component for positive disease outcomes for children with IBD, and has great potential to lessen the disease burden and its sequelae [7,21]. The health behaviors and processes that are required of the child with IBD while developing self-management skills may be cognitive, emotional, or social in nature [10,22]. In order to further define these a number of pediatric self-management frameworks were studied [10,21,23,24] in order to determine the core elements of self-management for review which were categorized as:

- Disease and treatment knowledge
- Self-regulation
- Adherence
- Communication
- Cognitive attributes

This synthesis review also highlighted that while skills in these core domains may be developed or achieved in isolation, in order to be attaining effective self-management these elements need to be addressed concurrently.

While the importance of self-management for children with IBD is increasingly recognized, it remains an emergent field and, subsequently, there is a dearth of assessment tools to measure these skills and attributes. Therefore, integrated approaches to address any gaps in children's self-management abilities are difficult to develop. Prior to developing interventions to maximize the potential for self-management it is vital to have standardized, reliable outcome measures to identify where the support is needed, and to establish the impact of any intervention through objective evaluation [25–27]. In order to achieve better disease outcomes for children with IBD it is important to understand the significance of self-management skills and attributes within the previously defined categories. The following review explores each self-management domain in detail and highlights a number of assessment tools that may be used to objectively measure them.

Relevant to all outcome measures identified, a number of factors should be considered when selecting assessment tools for use with children. These include whether items are age appropriate for the target population thereby considering the subject matter of included items, respondent burden, and readability. Respondent burden commonly refers to the number of items children are required to complete, and readability refers to the ease of comprehension a text should have so the target reader can understand it. Readability is frequently measured using the Flesch–Kincaid reading ease and school grade equivalence formulas [28,29]. The recommended reading ease score for children is stated as more than 70 out of a maximum of 100 [30], and the school grade score recommendations for low health literacy groups such as children should be below a grade five (age ten years) reading level [29].

## 2. Self-Management Factors

### 2.1. Disease and Treatment Knowledge

For children with IBD the acquisition of health knowledge regarding their own diagnosis, disease course, and treatment, is integral for their ongoing adherence and the development of self-management skills [31,32]. Acquiring knowledge of their disease and of treatment regimens are considered first steps in the process of developing health autonomy, and represent a concrete and tangible accomplishment [32,33]. Such knowl-

edge should be gradually developed to include, in concordance with the previous content synthesis [23,24,34,35]:

- Individual IBD knowledge; diagnosis, disease location and extent, surgical history, medication history
- General IBD knowledge
- Disease monitoring procedures and investigations
- Preventive health

Unfortunately, knowledge deficiencies among children with IBD regarding their own medical history are frequently reported, with the most common being: disease characteristics, surgical history, medication regimens, and how to contact their health care team [4,33,35,36]. Knowledge deficiencies such as these have consequences for pediatric self-management, as adults have reported that knowledge of disease processes, the role of medications, and of their treatment plan, were critical to being able to self-manage their condition [37]. In addition to understanding their own individual disease and treatment, it is vital to have a general understanding of their condition, the implications of their diagnosis, and the treatments available. Deficiencies are also evident across the pediatric literature in this domain, and are reported regarding drug and nutrition therapies, surgery, growth, and investigations [26,38–41].

Knowledge Outcome Measures

All children with IBD should have their disease and treatment knowledge levels assessed, as disease management may be adversely affected by gaps or misconceptions in either [42,43]. The most efficient way to evaluate knowledge is with an assessment tool that is appropriate for, and has been validated with, the target population. While the concept of knowledge is an abstract, subjective notion, questionnaires go some way towards assigning a value to what participants understand about a subject. When considering an assessment tool for use, it should be appropriate for the target population, as well as relevant to the most recent treatment modalities and knowledge.

Five studies were identified that measured general IBD knowledge in adults (Table 1); the Knowledge Questionnaire (KQ) [44], the Crohn's and Colitis Knowledge Score (CC-KNOW) [25], Inflammatory Bowel Disease Knowledge (IBD-KNOW) [45], the IBD-knowledge questionnaire Catalonia (QUECOMIICAT) [46] and a short questionnaire by Keegan et al. [47]. These assessment tools may be appropriate for adolescents but have not been validated in younger children and contain complex items, alongside topics such as pregnancy and smoking that may be inappropriate to ask young children.

There are three assessment tools specifically for measuring general disease and treatment knowledge in children with IBD [26,38], and four studies that measure individual disease and treatment information [4,33,35,48] (Table 1). Of those measuring general IBD knowledge in children, The Emma electronic quiz developed by Tung et al. [38] asked a series of twelve IBD questions that were automatically selected from a database of 185 items depending on the age and disease characteristics of participants, as well as four psychosocial questions. This tool was not found to be available for clinical use or research.

Haaland et al. [26] developed a tool called the 'IBD Knowledge Inventory Device' (IBD-KID), which was then re-developed following critical analysis [49] to a shorter, up to date version (IBD-KID2) [41]. IBD-KID2 has been validated among a number of population groups and has been shown to have appropriate readability and items that are generalizable between IBD clinical sub-types [50,51].

Of the four tools measuring individual disease and treatment information, three were informal and had not been validated, and one had undergone a process of validation. The non-validated assessment tools were presented in two formats; two studies utilized brief questionnaires regarding individual disease diagnosis, characteristics, and treatment [33,35] and the third study asked children and adolescents to fill out their own 'health passport' which could be carried at all times to provide information when required [4]. The fourth study presented a validated assessment tool called IBD KNOW-IT that asked a

series of questions aimed at establishing whether children knew about their own disease and treatment [48].

**Table 1.** Summary of knowledge assessment tool features to determine which is appropriate for target population.

| | First Author | Name | Year | Population | Items | Validity Tested | Reliability Tested | Readability Tested * | Transition Items ** |
|---|---|---|---|---|---|---|---|---|---|
| General IBD knowledge | Eaden [25] | CCKNOW | 1999 | Adults | 30 | Yes | Yes | Yes E 77.9, Gr 4.4 | Yes |
| | Jones [44] | KQ | 1993 | Adults | 9 | Yes | No | No | Yes |
| | Yoon [45] | IBD-KNOW | 2019 | Adults | 24 | Yes | Yes | Yes Gr 4.0 | Yes |
| | Keegan [47] | None | 2013 | Adults | 10 | Yes | Yes | No | Yes |
| | Casellas [46] | QUECOMIICAT | 2019 | Adults | 25 | Yes | Yes | No | Yes |
| | Tung [38] | Emma Quiz | 2015 | Children | 16 | No | No | No | No |
| | Haaland [26] | IBD-KID | 2014 | Children | 23 | Yes | Yes | Yes E 69, Gr 6.3 | No |
| | Vernon-Roberts [41] | IBD-KID2 | 2020 | Children | 15 | Yes | Yes | Yes E 77.2 | No |
| Individual knowledge | Fishman [33] | None | 2011 | Children | | No | No | No | No |
| | Gumidyala [35] | None | 2017 | Children | 12 | No | No | No | Yes |
| | Benchimol [4] | Health Passport | 2011 | Children | 28 | No | No | No | No |
| | Maddux [48] | IBD KNOW-IT | 2019 | Children | 23 | Yes | Yes | Yes E 75.5, Gr 6.3 | No |

* Readability: E = reading ease score/100, Gr = grade level equivalent; ** Relates to items specific to adult clinics, pregnancy, sex, recreational drugs, alcohol.

### 2.2. Self-Regulation

Self-regulation is considered an essential attribute for self-management and includes three skill components: self-monitoring, self-evaluation, and self-reinforcement [52,53]. These skills relate to proactive and reactive disease management, whereby actions are performed to manage a problem (example: IBD symptom self-monitoring), or respond to a change in condition (example: making lifestyle modifications or seeking medical help) [53].

#### 2.2.1. Symptom Self-Monitoring

Self-monitoring is a skill that enables children with IBD to track their symptoms in a structured way in order to promote reflection and awareness, and can augment communication of their disease state with the MDT. Tracking symptoms in this way is perceived as an efficient and cost-effective way for people to develop an awareness of their health state, but is also a means of documenting therapeutic benefit [54–57]. When used as a therapeutic tool, it has benefits over recall reports as it reduces the risk of recall bias, whereby symptoms become generalized. unless extreme events have occurred, which will skew the memory [58]. For adults with IBD, adherence to symptom self-monitoring is high, attributed to the fact that the data being collected are of personal importance [59]. In addition, commencement of self-monitoring has been reported to transform subsequent clinical encounters among adults with IBD who also considered that it enabled them to reflect on their disease [59].

### 2.2.2. Self-Evaluation

Self-evaluation for children with IBD is a skill that requires the assessment and reflection of their symptoms, and is a skill that should be learnt through education and reinforcement from the MDT and with parental support. When symptom self-monitoring is performed longitudinally it can provide information not just on their current disease state, but also provide retrospective data that can help them evaluate and reflect on changes in their condition. This enables them to recognize their symptom levels during remission, changes during periods of disease exacerbation, or assess treatment efficacy. Self-evaluation should have the discreet but beneficial effect of helping create an awareness of their own disease course and prompt the behavior of seeking timely medical help in the case of worsening symptoms [57]. Qualitative work carried out with adults with IBD reinforced that self-monitoring was considered a valuable way for patients to enhance consciousness of their health state, and to engage with their doctors [54].

### 2.2.3. Self-Reinforcement

The process of self-reinforcement is concerned with children with IBD learning to make decisions on what action is required when symptom changes are identified. This process emphasizes the importance of education and communication by the MDT. While a child can be taught to recognize symptom exacerbations, when this occurs in the pragmatic setting the child also needs to understand actionable instructions regarding what they should do and have decision support information available. Depending on the severity and nature of symptoms, it may be appropriate for them, or their parents, to call the GP, make contact with the MDT, or seek emergency help. Communication of their longitudinal symptoms to the appropriate party would enable a clinical evaluation to be made and the sharing of information relevant for this clinical evaluation is vital.

### 2.2.4. Self-Regulation Outcome Measures

In order for children with IBD to carry out symptom self-monitoring, evaluation, and reinforcement, an age appropriate and disease-specific tool is required that can provide symptom reports with clinical utility. Using a structured format for monitoring subjective variables, such as pain, well-being, and stool variables, can also quantify the disease burden for factors that are not readily observed but may help the MDT understand the child's perspective of symptoms [55,60].

The clinical assessment tools used by gastroenterologists for measuring disease activity are carried out using validated measures such as the Pediatric Crohn's Disease Activity Index (PCDAI) [61] and the Pediatric Ulcerative Colitis Activity Index (PUCAI) [62], or with a physician's global assessment (PGA) which is based on their clinical acumen. While these are not suitable to be used for child self-report due to their need for clinical data (PCDAI) and their complexity, adapted versions have been used with children with UC [57,63] and CD [57] that produce disease activity reports with good levels of crude agreement with clinician reports. No universal tool was identified in the literature that produced clinically relevant data via patient report that was universal and generalizable to both UC and CD.

A number of 'Patient Reported Outcome' (PRO) measures for children with IBD have been developed to enable them to report their symptoms. PROs are derived solely from patient input, provide feedback directly from the patient, and require no response interpretation by an observer [64]. The identified PROs were developed in conjunction with children with IBD and highlight those symptoms they consider most important. Two UC-specific PROs have been developed using similar methodologies: the TUMMY-UC [64], and the Daily Ulcerative Colitis signs and symptoms Scale (DUCS) [56]. These were developed using signs and symptoms derived from interviews with children with UC, with the purpose of providing patient reports that are intended to supplement the clinician completed PUCAI. The TUMMY-CD [65] is currently in development for children with CD using the same methodology. These PROs differ from clinical disease activity self-reports as they do not necessarily concentrate on, or reflect, the degree of inflammation or disease

activity but are designed to report perceived symptom burden [64]. These PRO measures consequently quantify a different concept to disease activity such as that measured by the PUCAI and PCDAI, and provide additional aspects of outcome measurement [64]. Once again, there was no tool universal to both CD and UC.

One clinical self-report tool for children with IBD called 'IBDnow' was identified that was developed using the same subjective symptom report sections as in the PCDAI and PUCAI [66]. This tool is presented as a series of picture and text Likert scales that are used to categorize pain, well-being, and stool variables such as blood, consistency, and frequency. IBDnow has a very simple format that enables children from a young age to provide symptom reports shown to have good agreement with clinicians and is generalizable for use among both clinical subtypes [66].

*2.3. Adherence*

Treatment regimens for IBD have been developed with proven efficacy and positive benefit-to-risk profiles, but in order to maximize outcomes children need to practice treatment adherence [67]. Adherence is a far more multifaceted phenomenon in childhood than in adulthood. A complex dyad exists between children and their parents over responsibility for treatment adherence [68]. Furthermore, a triadic partnership between the child, parents, and medical team must be in place to support multidimensional treatment components and the dynamic maturation of the child [69,70].

Adherence rates for children with IBD show great variability, with 16–80% reported as non-adherent to their prescribed regimen [11,71–73]. This variation may be a function of assessment method, patient age, and definition of the level of nonadherence [74]. Adherence to non-prescribed (over the counter) medications was significantly lower than to prescribed drugs [75], as was adherence to complementary medicines—the use of which has also been shown to also reduce adherence to prescribed treatments [73,76]. Adherence rates are higher for exclusive enteral nutrition (EEN) programs, which have been reported as 84% to 90% [77,78], although up to 33% of surveyed pediatric gastroenterologists reported non-adherence as the largest barrier to them using EEN [79]. Treatment adherence also includes attending scheduled clinic appointments, having investigations performed, and such factors as collecting new prescriptions to ensure ongoing drug adherence and performing recommended lifestyle changes. Adhering to clinic appointments has been shown to improve drug adherence, reduce the frequency of relapses, and improve remission rates [80–82]. Improved clinic attendance leads to stronger beliefs in the importance of medications, which in itself is a strong predictor of adherence [83].

### 2.3.1. Adherence and IBD Outcomes

For children with IBD medication adherence rates positively correlate with remission, and negatively correlate with disease severity [84,85]. Non-adherence is consistently associated with negative psychosocial outcomes such as reduced HRQoL, and increased health care utilization [86–89]. A number of risk factors for non-adherence have been identified in this population group; longer disease duration, high disease activity, greater age, use of herbal medications (a consequence of having too many drugs to take), having fewer follow-up appointments, and poorer parent-reported psychosocial HRQoL and family functioning [72,73,75,76,90,91].

### 2.3.2. Adherence Outcome Measures

An accurate assessment of medication non-adherence enables clinicians to view it as a diagnosable and treatable medical condition [92], and provides opportunities for education, to identify barriers, and to provide targeted interventions [93]. However, in order to design interventions to improve and maximize adherence, there first needs to be an understanding of which children are non-adherent and why [70]. No gold standard, validated adherence measure for children exists, and all identified techniques have been

proven to have limitations [94], therefore those direct, indirect, and subjective measures available should be examined for feasibility, accuracy, and limitations prior to their use.

The top three assessments reportedly used are subjective clinical interviews (with patient or parent), biological assay for drug markers (blood or urine), and a daily adherence diary, but there are many techniques available [95]:

- Patient or parent reports using interviews may be time consuming and subjective, and may overestimate adherence by up to 23% in adolescents with IBD when compared to objective measures [74,94,96–98]. The Medication Adherence Measure is a validated semi-structured interview that is widely used in pediatrics, and a correction factor for child and parent self-report data has been produced that should provide more accurate adherence rates from subjective reports [99].
- Bioassays for drug levels are validated, regulated, accurate, and objective. However, they are invasive, can be costly, and do not provide information on practical adherence such as doses missed [31,68,70,99,100].
- Daily adherence diaries are not validated and have a poor history of compliance, however, measuring the more universal concept of daily activities, incorporating medication taking, have better completion rates [97].
- Electronic medication monitoring devices can be used to track adherence to oral and inhaled drug regimens, thus providing objective, specific real time information on adherence [99]. This long-term monitoring method can reveal a spectrum of dosing problems, however, it relies on presumptive data on ingestion, is costly and prone to malfunctions [97,101].
- Pill counts involve totaling tablets (or liquid quantities) at two time intervals and comparing what is expected from the prescribed dosing regimen [70]. While this method is simple, feasible, and objective, it is also prone to inaccuracy and measures removal of the drugs from the container, not actual ingestion [70].
- Validated adherence scales are structured surveys that ask specific questions regarding adherence, with responses often measured using a Likert scale. None have been developed for children yet, but the most commonly used scale with adults is the Morisky scale [102], which has also been adapted for use with adults with IBD [103]. However, this scale measures barriers to adherence instead of nonadherence frequency, and may overestimate or undervalue adherence as items only account for daily medication regimens [98].
- A simple adherence visual analogue scale (VAS) provides a self-report method that is extremely quick to comprehend and complete. Studies comparing the Morisky scale to a simple VAS showed the VAS provides a more objective measure to quantify adherence [98].
- Pharmacy records regarding refill rates and the proportion of days covered by a filled prescription provide practical data on refill behaviors believed to correspond to medication taking. However, they do not directly estimate adherence and once again assume ingestion [31].
- The pediatric IBD disease activity indices (PUCAI and PCDAI) are frequently incorporated into adherence studies as a way of correlating symptoms with measured adherence.

None of these measurement techniques quantify exactly what each patient has taken and should be considered as measuring variables that are indicative of adherence rather than being measures of absolute medication use [104]. They do, however, provide an opportunity to triangulate multiple effective methods to provide accurate assessments of adherence [68,97].

### 2.4. Cognitive Attributes

Specific cognitive processes are inextricably linked to the development of self-management and health autonomy skills: patient activation, and self-efficacy.

### 2.4.1. Patient Activation

Patient activation is defined as when an individual demonstrates the necessary skills, knowledge, and motivation needed to self-manage their own health and participate in the decision-making process [105]. The defined stages of activation mark the progress of an individual from being a passive recipient of care that may be overwhelmed by the task, to them gaining knowledge and confidence in their skills, to eventually take action and perform relevant self-management behaviors [106]. Little has been published concerning patient activation in children, but it is has been associated with improved health outcomes in adults with IBD with those with higher activation levels being less likely to experience anxiety and depression, and more likely to be in remission [105]. Overall, adults who are active participants in their care are more likely to be adherent to their treatment regimen, engage in healthy behaviors, have lower health care utilization, and higher rates of accessing preventative care [106]. Higher patient activation is followed by improvements in self-management behaviors [106].

### 2.4.2. Self-Efficacy

Self-efficacy is a prerequisite for self-management, and has been shown to mediate the relationship between physical, psychological, and social functions with disease outcomes in a number of chronic diseases [107]. Self-efficacy relates to an individual's belief and confidence in their own capability to succeed in specific situations or to complete tasks [32,108], and high self-efficacy has been linked to successful health behavior change and better engagement with preventative health [107]. The importance of self-efficacy specifically in children with IBD is a nascent field, but during validation studies for self-efficacy assessment tools, higher levels were related to greater health care satisfaction, more frequent communication with the MDT, and higher IBD-related knowledge [35]. Among children with other chronic conditions those with higher self-efficacy levels reported better HRQoL, less depressive symptoms, and enhanced disease coping skills [7].

### 2.4.3. Cognitive Attribute Outcome Measures
### Patient Activation

The commonly used measure of patient activation among adults is the Patient Activation Measure (PAM) [109]. A shortened version developed by the same research group [110] has been used effectively in a cohort of children with IBD [111]. In addition, a parent version of the PAM (Parent-PAM) has been developed that may be utilized where an assessment is required of parents activation concerning their child's health [112].

### Self-Efficacy

Assessment tools for measuring self-efficacy in the adult IBD population were developed prior to those for children and adolescents. The IBD-Self Efficacy scale (IBD-SES) was first developed for adults with IBD in 2011 [107] and underwent further validation in 2016 [113]. This scale was adapted for use with adolescents to become the IBDSES-A [7], a scale validated for children with IBD aged twelve years and over [114]. The IBD-Yourself self-efficacy assessment tool has been validated for children aged fourteen and over, but is very long, with over 70 items, and therefore may be prohibitive for use in younger children [108].

### 2.5. Communication

Communication is integral to the development of self-management skills. When children are younger their communication with the MDT will be part of a triadic process between themselves, their parents, and the MDT, which will continue until their cognitive and emotional development eventuates in health autonomy [13,115]. It has been demonstrated that a more direct communication between physician and child during clinical encounters contributes to an improved relationship in terms of satisfaction with care, HRQoL, reduced worry, adherence to treatment, and better health outcomes [116–118]. However, when quantified, children have been shown to contribute only 4% of the time,

parents 35%, and doctors 61% [119]. Adults with IBD reporting poor communication with their clinician demonstrated a 19% higher risk of non-adherence than those reporting good communication [120].

Parents and the MDT can promote children's involvement by encouraging and inviting them to express their views, ask questions, and participate in discussions regarding their own health care [115]. The MDT can also provide education on how to respond to questions on health and illness by utilizing the child's individual and general IBD knowledge, a process which leads to navigational health literacy, a necessary attribute for good decision-making [34,116,121].

Communication Outcome Measures

In order to help children with IBD communicate with their family and the MDT about their condition, structured methods of reporting may be considered beneficial. While encouraging children to communicate regarding their current and retrospective disease state is indisputably important, defining and describing these concepts can be challenging for many children. Education regarding self-regulation, and the provision of self-monitoring tools, may help clarify these issues for the child and enable their dialogue to be of value in the clinical setting [54]. There is, therefore, an ongoing need to provide practical tools to support self-regulation that may help address communication gaps, and will enable a shared-care approach to IBD management [122].

Measuring communication skills among children with IBD should be done in subjective and objective ways. Subjectively, ongoing attention to a child's level of interaction with the MDT during consultations will identify areas for encouragement and support. Prompting children to compile a list of questions to ask during clinical appointments, and to report their own treatment, is a simple way to begin when encouraging the development of self-management skills. Following education regarding self-regulation and disease knowledge, they can be encouraged to report their symptoms and join clinical discussions regarding their ongoing management. Providing children with objective measures for symptom reports, such as the IBDnow tool [66] can facilitate a structured method of communicating their disease state in terms of their symptom burden, and longitudinal reflection can encourage them to establish flare recognition and evaluate treatment efficacy. Children keeping a diary of their treatment, symptoms, and questions, thereby encouraging a habit that will be beneficial throughout their disease course into adulthood, may combine such subjective and objective methods.

*2.6. Self-Management Skills*

This review has identified those processes and behaviors considered integral to the development of self-management skills for children with IBD: knowledge, self-regulation, adherence, cognitive attributes, and communication. Concurrent development of these self-management skills contributes to optimal disease outcomes throughout childhood, and maximizes the chance of a successful transition from the pediatric to adult health care team [120]. It is therefore appropriate to identify those methods available for assessing overall self-management skills relating to all domains in order to provide targeted interventions for those skills requiring additional support.

A number of approaches to measuring self-management in children have been suggested including measuring the allocation of responsibility for health care tasks between children and their parents, or by quantifying the level of self-management skills with a numeric score that can be measured longitudinally to determine if skills are increasing [123]. The appropriate assessment of self-management skills in the research setting can quantify efficacy of targeted interventions, and in the clinical setting may identify children with IBD at risk of sub-optimal health autonomy who could benefit from MDT input as targeted education and interventions.

Self-Management Outcome Measures

Studies addressing self-management interventions for adults with IBD used the Health Education Impact Questionnaire that measures skills, behaviors, and cognitive aspects across a number of domains [124]. This tool is not IBD-specific, and has not been validated among children. An initial search for self-management assessment tools specific for children with IBD revealed a wide range of approaches, subjects, and formats with some tools labelled for self-management instead pertaining to transition readiness, knowledge, self-efficacy, or education. A number of tools have also not been validated. Given the dearth of specific tools, it was considered pertinent to perform a review of those identified to ascertain if any were appropriate for use in the whole pediatric IBD population for the assessment of practical self-management skills.

Nine tools were identified through literature review as being related to self-management assessment (Table 2). The two simplest tools are non-validated IBD transition checklists that have been developed from the literature, expert opinion, and anecdotal evidence and provide a guide for age expectations for the development of particular skills [5,125]. Two further tools were presented that could be used by children with IBD to report whether self-management tasks could be performed by the participants on their own, or with varying levels of help [32,126]. The tool presented by Whitfield et al. [126] has been devised by the ImproveCareNow IBD network in the U.S [127] and is included in their self-management manual [128], but has not been validated. Four tools were identified that were generic but been validated among children with IBD or long term gastrointestinal conditions; StarX [129,130], UNX-Transition Scale [131], Transition-Q [132], and a tool by Williams et al. [133]. Three of these generic tools contained items that were related specifically to transition and concerns for adolescents which could be deemed inappropriate for younger children with IBD who may be learning self-management (examples: adult clinics, smoking, drugs, pregnancy, sex, and alcohol), or developed in the US and contained items relating to health insurance relevant only to that setting. The final tool was a practical self-management skills assessment tool called IBD-STAR that is validated for children with IBD over the age of eight years [134]. IBD-STAR assigns scores for a number of practical self-management tasks depending on allocation of responsibility by the child with IBD, and has been shown to produce reports from participants that are in line with the age expectations in the checklists previously identified [5,125].

**Table 2.** Summary of self-management skills assessment tool features to determine which is appropriate for target population.

| First Author | Name of Tool | Year | Origin | Topic | Age Range | Items | Validity Tested | Reliability Tested | Readability Tested | Insurance * | Transition ** |
|---|---|---|---|---|---|---|---|---|---|---|---|
| Hait [5] | None | 2006 | US | IBD T | 11–23 | 17 | No | No | No | Yes | No |
| Fishman [32] | None | 2010 | US | IBD SM, T | 16–18 | 19 | No | No | No | No | No |
| Whitfield [126] | None | 2015 | US | IBD SM | 10–21 | 23 | No | No | No | No | Yes |
| NASPGHAN [125] | None | 2010 | US, CAN | IBD T | 12–17 | 27 | No | No | No | Yes | Yes |
| Klassen [132] | Transition-Q | 2014 | CAN | SM T | 12–18 | 14 | Yes | Yes | Yes Gr 4.4 | No | No |
| Ferris [130] | StarX | 2015 | US | SM T | 12–25 | 18 | Yes | Yes | Yes Gr 4.4 | No | Yes |
| Ferris [131] | UNX-Transition Scale | 2012 | US | SM T | 12–20 | 33 | Yes | Yes | No | Yes | Yes |
| Williams [133] | None | 2010 | CAN | SM T | 11–18 | 21 | Yes | Yes | Yes Gr 4.9 | Yes | Yes |
| Vernon-Roberts [134] | IBD-STAR | 2020 | NZ | IBD SM | 10–18 | 18 | Yes | Yes | Yes E 91, Gr 3.1 | No | No |

Country of origin: US = United States, CAN = Canada, NZ = New Zealand; Topics: IBD, T = transition, SM = self-management; readability: E = reading ease score/100, Gr = grade level equivalent; * relates to items specific to health insurance, ** relates to items specific to adult clinics, pregnancy, sex, recreational drugs, alcohol.

### 3. Conclusions

This review has outlined the multi-faceted nature of self-management and the importance of providing a cohesive approach to the essential processes involved; knowledge, communication, adherence, self-regulation, and cognitive factors (Figure 1).

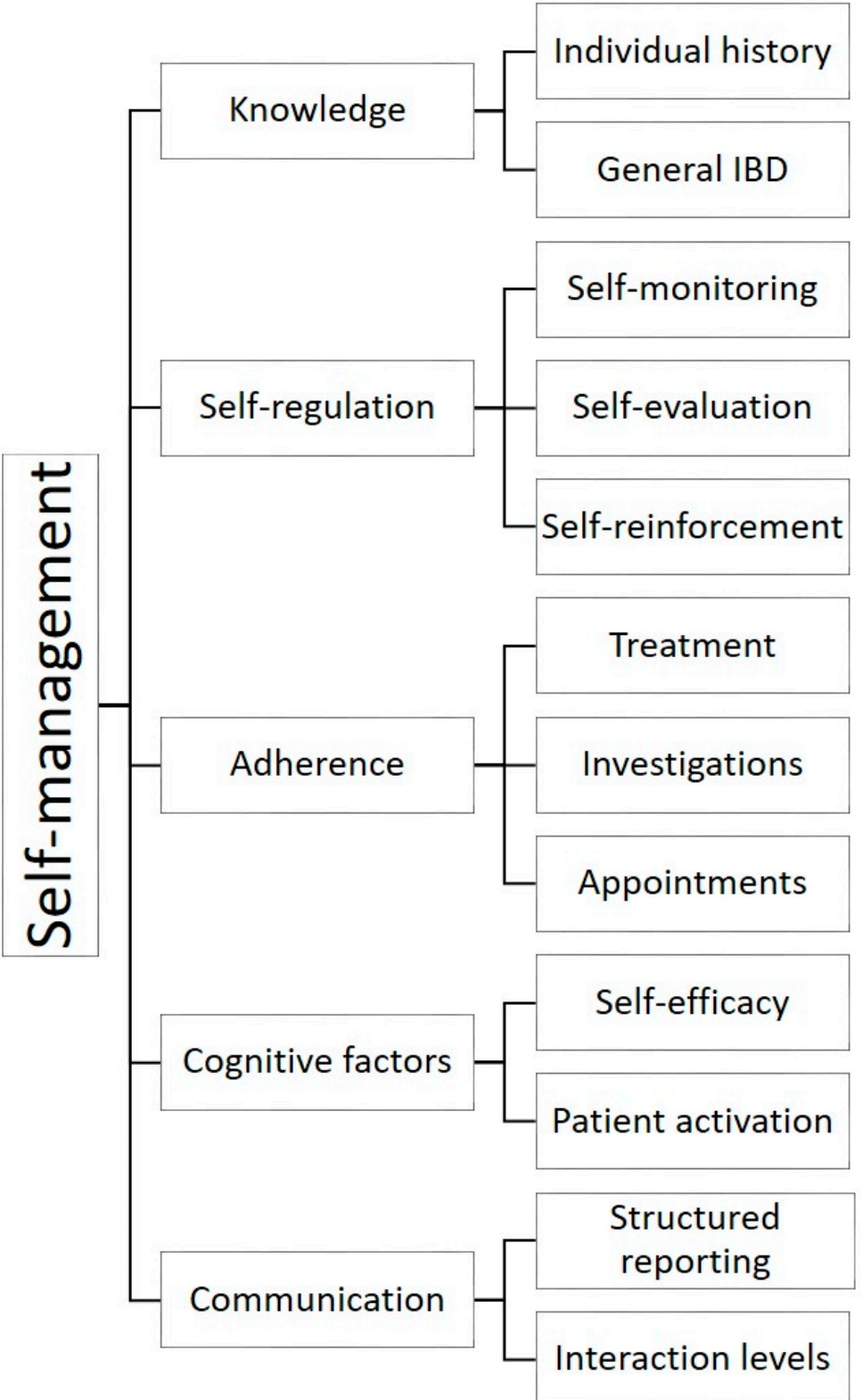

**Figure 1.** Factors relating to the development of self-management skills.

The development or identification of interventions that address all of these inter-related components is vital in order to improve outcomes for children with IBD, and efficacy should be assessed using validated outcome measures. Clinicians and the MDT should recommend practical self-management activities to children with IBD and their parents during clinical encounters, and children routinely assessed with targeted outcome measures to identify where additional support may be beneficial.

**Author Contributions:** Conceptualization, A.V.-R., R.B.G. and A.S.D.; writing—original draft preparation, A.V.-R.; writing—review and editing, A.V.-R., R.B.G., A.S.D. All authors have read and agreed to the published version of the manuscript.

**Funding:** This research received no external funding.

**Conflicts of Interest:** The authors declare no conflict of interest.

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
