# Peer review of "Overview of Self-Management Skills and Associated Assessment Tools for Children with Inflammatory Bowel Disease"

_gastrointestdisord, doi:10.3390/gidisord3020007_

Round 1

Reviewer 1 Report

General comments:

The authors reviewed investigated self-management skills and associated assessment tools for children with inflammatory bowel disease.

The article was written each self-management domain a lot. But this review could not “identify assessment tools that may be used to objectively measure them”. Only small number of readers could imagine what kind of quoted tool.

Specific comments:

  1. Please include specific examples (tasks and figures) of the tool instead of many sentences. The focus should be on those with a lot of reports.
  2. Please describe the important points in children differently from adults.
  3. Please match the formats of Table 1 and Table 2.

Reviewer 2 Report

  • Could reasoning about adherence lead to one therapy over another (e.g., intravenous infliximab instead of subcutaneous adalimumab)?

  • Please provide a few details about the methodology of the search of this narrative review

  • The last paragraph is more a conclusion than a discussion: please rename it

Round 2

Reviewer 1 Report

Thank you for your response. I had no additional comments.